# Open Partial Laryngectomies: History of Laryngeal Cancer Surgery

**DOI:** 10.3390/jcm11185352

**Published:** 2022-09-12

**Authors:** Stéphane Hans, Robin Baudouin, Marta P. Circiu, Florent Couineau, Quentin Lisan, Lise Crevier-Buchman, Jérôme R. Lechien

**Affiliations:** 1Department of Otorhinolaryngology and Head and Neck Surgery, Foch Hospital, School of Medicine, UFR Simone Veil, Université Versailles Saint-Quentin-en-Yvelines (Paris Saclay University), 92150 Suresnes, France; 2Division of Laryngology and Broncho-Esophagology, EpiCURA Hospital, UMONS Research Institute for Health Sciences and Technology, University of Mons (UMons), 7000 Mons, Belgium; 3Polyclinic of Poitiers—Elsan, 86000 Poitiers, France

**Keywords:** larynx, laryngeal, cancer, partial laryngectomy, otolaryngology, head neck, surgery, history

## Abstract

Historically, surgery was the first-choice therapy for early, intermediate and advanced laryngeal squamous cell carcinoma (LSCC). Partial laryngeal surgery has evolved in recent decades and was influenced by many historic events and the development of new technologies. Partial laryngectomies may be performed by open, endoscopic or transoral robotic approaches. In this historic paper, we describe the evolution of open partial laryngectomy techniques, indications and surgical outcomes. Since the first partial laryngectomy in 1788, many U.S., U.K. and European surgeons, including Henry Sands, Jacob da Silva Solis-Cohen and Theodor Billroth, performed this surgical procedure under local anesthesia for tuberculosis, cancer or syphilis. Partial laryngectomy gained reputation in the medical community in 1888 due to the laryngeal cancer and death of the prince of Prussia, Frederick III. Frederick III’s death represented the turning point in the history of partial laryngectomies, calling attention to the importance of semiotics, biopsy and early diagnosis in laryngeal cancers. Hemi-laryngectomy was indicated/proposed for lateral laryngeal tumors, while thyrotomy was indicated for cancers of the middle part of the vocal fold. The second landmark in the history of partial laryngectomies was the discovery of cocaine, novocaine and adrenaline and the related development of local anesthetic techniques, which, together with the epidemiological and hygienic advances of the 19th century, allowed for better perioperative outcomes. General anesthesia was introduced in the second part of the 20th century and further improved the surgical outcomes. The diagnosis of laryngeal cancer was improved with the development of X-rays and direct laryngoscopies. The 20th century was characterized by the development and improvement of vertical partial laryngectomy procedures and the development of horizontal partial laryngectomies for both supraglottic and glottic regions. The history and the evolution of these techniques are discussed in the present historical paper.

## 1. Introduction

Head and neck squamous cell carcinoma is the sixth most common adult cancer worldwide, accounting for 5.3% of all cancers [1]. Among head and neck cancers, laryngeal squamous cell carcinoma (LSCC) is the second most prevalent carcinoma, corresponding to 211,000 new cases and 126,000 deaths per year worldwide [2,3]. The treatment of LSCC depends on tumor location, stage and patient comorbidities. Historically, surgery was the first-choice therapy for early, intermediate and advanced laryngeal squamous cell carcinoma (LSCC). Partial laryngeal surgery evolved over recent decades and was influenced by several historically important cases as well as by technological events. At present, partial laryngectomies may be performed through open, endoscopic or transoral robotic approaches. In this paper, we describe the evolution of the techniques, indications and surgical outcomes of partial laryngectomies.

The existing literature describes dozens of types of open partial laryngectomies. In our work, we choose to focus on a selection of publications that played an important role in illustrating two therapeutic extremes: the transoral approach and total laryngectomy. We also provide historical examples of partial laryngectomies to support examples. We learned about these delicate surgeries in the “School of Laënnec” in contact with professors Henri Laccourreye, Daniel Brasnu and Ollivier Laccourreye. Finally, we use the European Laryngological Society (ELS) classification for open partial horizontal laryngectomies (OPHL) [4].

## 2. The Pioneers

The first partial laryngectomy was carried out by the French surgeon Philippe-Jean Pelletan (1747–1829) in 1788 [5]. Philippe-Jean Pelletan proceeded to perform a midline thyrotomy to remove a laryngeal piece. The first partial laryngectomy performed for a laryngeal cancer was reported in 1863 by Henry B. Sands (1830–1888), who was a faculty surgeon at New York University [6]. The surgery consisted of a laryngeal fissure and extirpation of a laryngeal tumor in a patient who died two years after the surgery without evidence of recurrence [6]. In 1867, Jacob da Silva Solis-Cohen (1838–1927), a laryngologist from Philadelphia, published the first long-term follow-up (20 years) of a patient who underwent a median thyrotomy for a presumed laryngeal cancer. In that paper, the disease was still controlled 20 years after the surgery [7].

The first open hemi-laryngectomy was carried out in 1878 by Theodor Billroth (1829–1894), a few years after the first total laryngectomy (1873) [8]. At this time, total or partial laryngectomies were performed for three main types of chronic ulcerative laryngitis: tuberculosis, cancer and syphilis. The surgeries were not preceded by a biopsy because surgeons considered that biopsy increased the growth of cancer [9].

## 3. The Prince of Prussia

In January 1887, the prince of Prussia, Frederick III (1831–1888), was diagnosed with a laryngeal cancer, which was a milestone in the diagnosis and the evolution of the surgical treatment of laryngeal cancer [10,11]. The heir of the German Empire requested the expertise of Karl Gerhardt, who reported a thickening of the left vocal fold and prescribed thermal therapy at Bad Ems. On 15 May 1887, the doctor observed a left vocal fold movement impairment. The heir was examined by Ernst Von Bergmann (1836–1907), who clinically confirmed the possibility of laryngeal malignancy and proposed a laryngofissure. The family of the heir, especially his wife, the princess Victoria, requested a second opinion from Morell Mackenzie (1837–1892), an English laryngologist who suspected syphilis and performed a laryngeal biopsy. The pathological analysis was carried out by Rudolf Virchow (1821–1902), one of the leading physicians to Frederick III, who diagnosed “pachydermia”. Based on this examination, the prince’s relatives opposed the laryngectomy. Meanwhile, Emperor Guillaume I died, and Frederick III became Emperor. The reign of Frederick III lasted 99 days, and he died from the evolution of his laryngeal cancer on 15 June 1888. Before his death, physicians performed a tracheotomy and several additional biopsies that were all negative. According to the clinical evolution of the laryngeal disease, Rudolf Virchow and Heinrich W. Waldeyer (1836–1921) carried out an autopsy that supported the laryngeal cancer diagnosis. Because Frederick III was a politically important and valued person, the origin of his death led to debates and reflections regarding the importance of semiotics, biopsy and the early diagnosis of laryngeal cancer in the consideration of partial laryngectomies. The spread of these debates in Europe reinforced the importance of partial laryngectomies and several procedures were carried out in the U.K. (1894) and France (1895) by Sir Felix Semon (1849–1921) and Emile Jean Moure (1855–1941), respectively [8]. At this time, the discussions were focused on organ preservation via two main techniques: thyrotomy and Billroth’s hemi-laryngectomy. However, the indications remained unclear. The median thyrotomy was proposed for cancers of the middle part of the vocal cord without both anterior commissure invasion and laryngeal dysmotility. The lateral laryngeal tumors required hemi-laryngectomy. One of the first studies was published in 1897 by John Sendziak [12,13]. In this study, the author reported that the postoperative mortality rate and 3-year overall survival of midline thyrotomies (*n* = 88) were 9.8% and 8.7%, respectively, while total laryngectomies (*n* = 188) were associated with a postoperative mortality rate and overall survival of 44.7% and 5.85%, respectively [13]. At that time, many surgeons proposed palliative tracheotomy rather than partial or total laryngectomies.

## 4. The First Part of the 20th Century

### 4.1. The Influence of Hygiene and Anesthesia Development

At the end of the 19th century, patient anesthetization was performed with ether or chloroform. In the postoperative period, the prevalence of pneumonia, wound infections and other complications was high. The discovery of cocaine, novocaine and adrenaline led to the development of better local anesthetic techniques, and many laryngeal surgeries were, therefore, carried out under local anesthesia. At the same time, European progress in infectiology, hospital hygiene and epidemiology, achieved through the works of Ignace Semmelweis (1818–1865), Louis Pasteur (1822–1895) and Joseph Lister (1827–1912), led to better conditions in the operating room and better postoperative outcomes.

### 4.2. The Influence of X-rays

The discovery of X-rays in 1895 by Wilhelm Röntgen (1845–1923) is another event that significantly influenced partial laryngeal surgery. This discovery provided laryngologists and surgeons with a new imaging and exploration technique for the larynx. They performed profile images or frontal tomograms, allowing the visualization and characterization of both the epiglottis and the ventricle tumors. X-ray development improved the indications of partial laryngectomies because for more than 50 years, the surgical indications were based on palpation, indirect laryngoscopy and in-office biopsy under local anesthesia with cocaine.

### 4.3. Classifications and Laryngoscopy

The study of the location and extensions of laryngeal cancers developed during the second part of the 19th and throughout the 20th centuries. Emile Isambert (1827–1876) and Maurice Krishaber (1836–1883) reported two types of laryngeal cancers: the ‘*extrinsic laryngeal cancer*’, with cervical nodes and poor prognosis, and the ‘*intrinsic laryngeal cancer*’, without neck extension and better prognosis [14]. A few decades after this first classification, the laryngeal cancers were classified into subglottic, glottic or supraglottic according to the studies of Henri Rouvière (1876–1952) and others [15,16]. Laryngeal dissections led to a better understanding of the weak and resistant regions of the larynx and, therefore, the potential extension pathways of cancer [17]. At this time, the laryngeal examination was performed with the Garcia laryngeal mirror [18]. In 1895, Alfred Kirstein (1863–1922) performed the first direct laryngoscopy a few years before Gustav Killian (1860–1921), who also developed a laryngoscopy procedure (Figure 1) [19,20,21].

### 4.4. Vertical Partial Surgery

Many vertical partial surgery techniques appeared in the first part of the 20th century and were popularized by Chevalier Jackson (1865–1958), St. Clair Thomson (1859–1943) and others [16,22]. These approaches were developed to treat laryngeal cancer without requiring a total laryngectomy, which was seen as a mutilating approach. Three types of partial laryngectomy developed over time.

The first type consisted of thyrotomy with uni- or bilateral cordectomy, with potential extension to the arytenoid cartilage.

The second type was the vertical partial laryngectomy (fronto-lateral type) (Figure 2), which removed a part of the thyroid cartilage in the midline without cricoid cartilage resection. Described in 1956 by Leroux-Robert (1907–1998) [23], the fronto-lateral laryngectomy was indicated for cT1a tumors. This intervention included a monobloc resection of the affected vocal fold, the anterior commissure, the anterior part of the contralateral vocal fold and a part of the thyroid cartilage. To improve vocal function, glottic reconstruction was performed with repair of the false vocal fold [24]. To date, indications of fronto-lateral laryngectomy have decreased thanks to the development of transoral laser microsurgery. However, we currently use this vertical partial laryngectomy for tumors of the middle third of the vocal fold without involvement of the anterior commissure not exposed by TLM.

The third type was hemi-laryngectomy, which was initially described by Billroth in 1878. This historical approach consisted of a median thyrotomy and the resection of the hemi-thyroid and hemi-cricoid laryngeal region affected by the tumor. Anything less than this resection was reported as a laryngofissure [9,16].

The postoperative outcomes of a vertical partial laryngectomy depended on the tumor location. Authors reported adequate outcomes for tumors limited to the middle third of the mobile vocal cord. They also noted that the closer the tumor was to the front or back of the vocal cord, the lower the chances of a successful surgery. [16]. The procedure was performed in Europe throughout the first part of the 20th century with the specification that the fixation of the arytenoid cartilage was a contraindication to partial laryngectomies [25,26]. The failure rate of vertical partial laryngectomies ranged from 2% to 18% for cT1 and from 4% to 24% for T2 laryngeal cancer. The failure rate was more than 40% if the vocal fold was fixed [16]. Since the spread of partial laryngectomy approaches in Europe in the first part of the 20th century, many modified approaches were reported with adequate outcomes [27]. However, the limitations of these techniques lay in the concept of relative independence of the two hemi-larynxes, which led to the advent of the horizontal laryngectomies.

## 5. The Second Part of the 20th Century

The second part of the 20th century included the development of horizontal partial laryngectomies, such as supraglottic laryngectomy (OPHL type I), glottic partial horizontal surgery (OPHL type IIa), and partial horizontal surgery of both glottic and supraglottic regions (OPHL type IIb) (Figure 3) [4]. We do not discuss supratracheal laryngectomies (OPHL type III) because they most often require a tracheotomy. In our opinion, the aim of a partial laryngectomy is to obtain the same local control as that obtained through a total laryngectomy, with oral swallowing (without a feeding tube), phonation and breathing without a tracheostomy.

### 5.1. The Horizontal Partial Supraglottic Laryngectomy (OPHL Type I)

The concept of the horizontal partial laryngectomy was based on Hajek’s anatomical studies of the lymph node, which were developed through the studies of Henri Rouvière and Francois Baclesse (1896–1967) [28,29]. Initially, the first practical approaches for the resection of tumors of the supraglottic larynx and lateral pharyngeal wall were published by Wilfred Trotter (1872–1939) in a 1913 edition of *The Lancet* [30]. The techniques were then developed by a number of French surgeons [16], while the Uruguayan surgeon Justo M. Alonso (1886–1974) developed a voice-sparing supraglottic laryngectomy in 1947 [31]. Alonso’s techniques were modified and spread worldwide by important laryngeal surgeons of the modern era, including Max L. Som (1904–1990), Joseph H. Ogura (1915–1983) and Jean Leroux-Robert [22,23] in France and Ettore Bocca in Italy [32]. The procedures evolved with the extension of the supraglottic laryngectomy to the base of the tongue, the arytenoid cartilage and the piriform sinus [4,33]. Preserving the mobility of one arytenoid unit was an important issue for supraglottic partial surgeries, while the postoperative course required a transient tracheostomy, feeding tube and hospital follow-up for nearly three weeks. Currently, these surgeries are still carried out but the development of transoral microsurgery or robotic approaches (early stages) or chemoradiotherapy (advanced stages) have reduced the indication of open partial laryngectomy.

### 5.2. The Horizontal Partial Glottic Laryngectomy (OPHL Type IIa)

The horizontal partial glottic surgery included several procedures classified according to the resection of the thyroid cartilage.

The first approach was developed by Tucker et al. (U.S.) [34] and consisted of a vertical resection of the thyroid cartilage preserving the posterior part of the thyroid wings. This approach was indicated for cT1–T2 laryngeal cancers with normal mobility of the vocal folds and was particularly interesting for tumors with an involvement of the anterior commissure.The second approach is the supracricoid laryngectomy with crico-hyoido-epiglottopexy (CHEP) (Figure 4) [35,36,37]. In this procedure, the surgeon removes the thyroid cartilage; conserves the cricoid cartilage, hyoid bone and at least one arytenoid unit; and performs an epiglottopexy through a suture between the cricoid and hyoid (sure 4). At least one cricoarytenoid unit (cricoid, arytenoid, cricoarytenoid joint/muscles and the associated recurrent nerve) must be preserved for the functioning of the laryngeal sphincter. The technique was proposed by Majer and Rieder in 1957 and spread by Jean-Jacques Piquet et al. (1974), Henri Laccourreye and Daniel Brasnu under the name supracricoid laryngectomy with reconstruction by CHEP [37]. The indications were cT1–T2 tumors and some selected cT3 tumors with fixation of the vocal cord and normal mobility of the arytenoid cartilage. This approach was particularly interesting for tumors with involvement of the anterior commissure.

The postoperative evolution of these two types of surgery required a transient tracheostomy, feeding tube and hospital stay of 3–4 weeks. Such procedures were associated with adequate local and regional control rates for selected vocal fold tumors [34,36,37].

### 5.3. The Horizontal Partial Procedures for Glottic and Supraglottic Regions

In this group of partial laryngectomies, two main procedures may be described, depending on the type of resection of the thyroid cartilage.

The first approach consisted of a partial resection of the thyroid cartilage and was known as the three-quarter laryngectomy. This approach was developed by Bocca et al. in Italy (1971) [38] and Dedo in the U.S. (1975) [39].Described in 1971 by Labayle and Bismuth [40], the second procedure consisted of a complete resection of the thyroid cartilage followed by a reconstruction through a crico-hyoido-pexy. The approach was called ‘supracricoid laryngectomy with reconstruction by crico-hyoido-pexy’ by Laccourreye and Brasnu [41].

Currently, both surgical approaches of the supraglottic and glottic regions are rarely carried out due to the increase in organ preservation protocols indicating chemotherapy and radiotherapy.

At the end of the 20th century, the 5-year local control rate and the 5-year laryngeal preservation rate of vocal cord cancers were 92–95% and 95–100%, respectively. For supraglottic cancers, the 5-year local control rate and the 5-year laryngeal preservation rate were 92–94% and 92–95%, respectively [42].

Many studies have demonstrated that vocal quality after a partial laryngectomy is related to the extent of the resection and the reconstruction methods. The spoken voice is called the “neoglottic substitution voice” after a partial vertical laryngectomy [24,43] and the “neolaryngeal substitution voice” after a supracricoid laryngectomy (OPHL type IIa/IIb) [44,45,46]. For open partial laryngectomies, the voice characteristics will depend on (i) the shape and nature of the remaining structures and the amount of effort and adaptation required to reach a vibrating neoglottic/neolaryngeal voicing sphincter; (ii) collateral constraints of external surgeries, such as a transient tracheostomy or nasogastric tube, that modify and lengthen the dynamics of vocal recovery; and (iii) for endoscopic surgeries, whether the physiological shape of the phonatory glottis is not modified. Thus, vocal rehabilitation aims to improve the closure of the neoglottis and the quality of the mucosal vibration with voice tone improvement (see *Laryngeal Cancer Surgery—Part II*). After a partial laryngectomy, voice rehabilitation is long and requires the patient’s effort. In the case of vertical partial laryngectomies, many techniques (e.g., flap) have been developed to improve the voice [24,43]. According to our experience and the data in the literature, vocal progress is achievable even several months after a partial laryngectomy [45,46].

### 5.4. Evolution of Partial Laryngectomies over the Last Thirty Years

In the past thirty years, there has been a major transformation in the way we treat LSCC, including a decline in the use of open surgery as first-line treatment for a certain proportion of these tumors [47,48,49]. This evolution was made possible by several factors.

First, the incidence of LSCCs has decreased in most developed countries, partially as a result of public health agencies’ efforts to decrease tobacco consumption. Second, advances in chemotherapy and radiation therapy (RT) have led to highly effective non-surgical regimens for patients with advanced laryngeal cancers, with the added advantage of laryngeal preservation in many cases. The Veterans Administration Study published in 1991 established the fact that the response to a neoadjuvant CT scan predicts the response of a tumor to RT. Patients with advanced tumors that responded either partially or completely to CT were treated with RT, and total laryngectomy was reserved for non-responders. This made it possible to preserve the larynx in a significant number of patients with locally advanced laryngeal cancer, while achieving local control and overall survival results equivalent to those achieved with initial total laryngectomy. By 2003, the results of the RTOG 93-11 trial, utilizing CCRT as initial treatment, were published, demonstrating a higher rate of laryngeal preservation with this protocol. Surgery was reserved for treatment failures. This concept changed the paradigm for management of advanced laryngeal cancer, greatly reducing the number of laryngectomies performed. While partial supracricoid laryngectomy has been employed for selected patients, total laryngectomy is the usual procedure for salvage *or* failure after nonsurgical treatment.

Third, technological advances with widespread availability, such as operating microscopes; endoscopes; lasers; image-guided surgery; and more recently, robotics, are transforming our surgical approaches, with transoral minimally invasive techniques greatly improving the postoperative course and functional outcomes for selected tumors (see *Laryngeal Cancer Surgery—Part II*).

### 5.5. Current Indications for Partial Laryngectomies

From our experience and recently published articles, the current indications for partial laryngectomies are laryngeal tumors with inadequate transoral exposure, certain tumors of the anterior commissure with vertical development (see *Laryngeal Cancer Surgery—Part II*) and selected salvage LSCCs, after radiation therapy failure. However, the experience of many contemporary surgeons with partial laryngectomy is quite limited. For the treatment of localized RT-resistant laryngeal cancer, the surgeon must be perfectly familiar with the type of extension of the tumor as well as with the indications for partial laryngectomies. Nevertheless, partial laryngectomy should be used with caution in patients requiring salvage surgical therapy for a recurrent or persistent laryngeal tumor. Recurrences after RT tend to be submucosal and difficult to evaluate. The only type of partial laryngectomies reported in the literature are the supracricoid partial laryngectomies [50,51]. In a systematic review, De Virgilio reported eleven papers (251 patients from 1990 to December 2017) with 2-year local control and 5-year overall survival rates of 92 and 70%, respectively. The larynx preservation rate was 85.2%. The decannulation rate was 92.1%, and the swallowing recovery rate was 96.5% (PEG dependance and the aspiration pneumonia rate were 3.5% and 6.4%, respectively) [51].

## 6. Conclusions

Throughout the 20th century, many surgical techniques were developed to avoid total laryngectomies. These procedures were developed through progress in anesthesiology, hygiene and infectiology, as well as due to the expertise of many laryngologists. These open partial laryngectomies led to the preservation of many larynxes around the world while reporting adequate oncological and functional outcomes (voice, breathing and swallowing). Although promising/beneficial for the patient, these approaches have some limitations. First, they require a transient tracheostomy and the use of feeding tubes and are associated with long hospital stays. The voice and swallowing rehabilitation program is long and requires patient motivation. Moreover, the surgery is associated with aesthetic sequelae, such as cervical scars.

In the past thirty years, major modifications in the way we treat LSCC, due to the decreasing incidence of tobacco-related LSCC in the West, as well as advances in technology (Transoral Laser Microsurgery, Transoral Robotic Surgery) and medical oncology, have led to a decline in the use of partial laryngectomy as a first-line treatment of LSCC.

## 7. In Tribute

This paper is a tribute to Professor Henri Laccourreye, Head of the ENT and Head and Neck Surgery Department at Laënnec Hospital from 1978 to 1992; we also thank Associate Professor Ollivier Laccourreye and Madeleine Ménard.

This paper is also a tribute to Professor Daniel Brasnu, Head of ENT and Head and Neck Surgery at Hôpital Laënnec from 1992 to 2000, and then Head of ENT and Head and Neck Surgery at Hôpital Européen Georges Pompidou from 2000 to 2015.

## Figures and Tables

**Figure 1 jcm-11-05352-f001:**
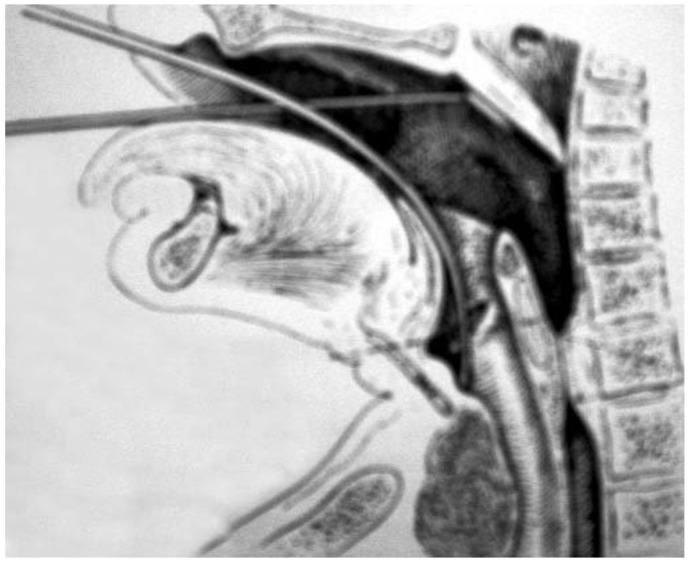
Killian laryngoscopy technique. This picture shows the technique of direct laryngoscopy developed by Killian.

**Figure 2 jcm-11-05352-f002:**
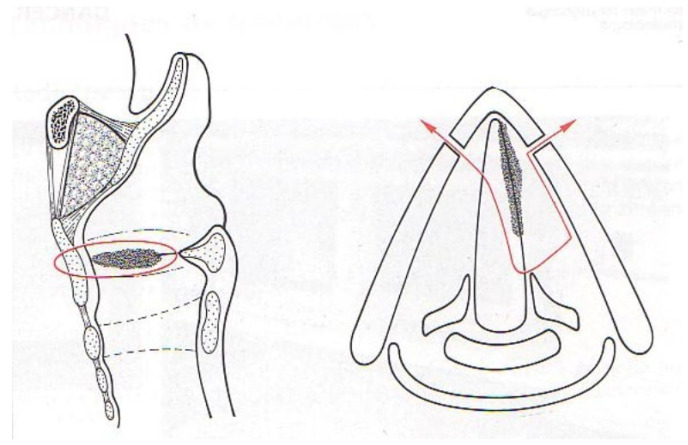
Fronto-lateral laryngectomy. Monobloc resection, removing a vertical fragment of the thyroid cartilage, of the entire vocal cord, of the anterior commissure and of the anterior part of the contralateral vocal cord.

**Figure 3 jcm-11-05352-f003:**
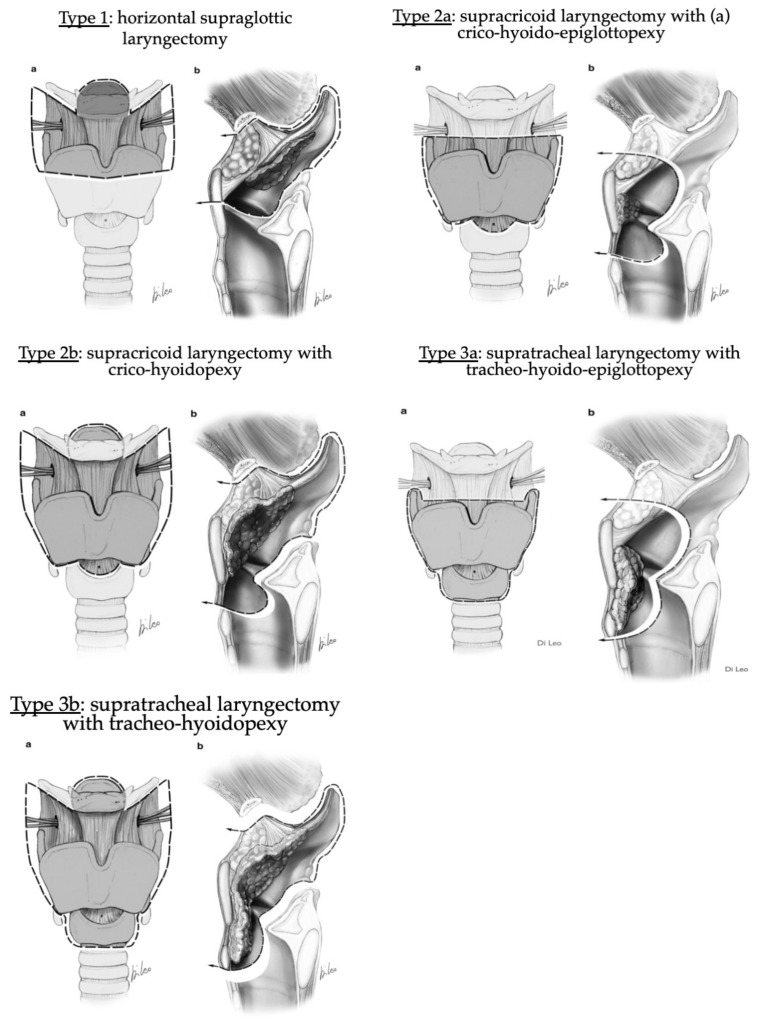
European Laryngological Society Classification of Open Partial Laryngectomies. Authors received the authorization of Pr. Marc Remacle to re-use the picture [4].

**Figure 4 jcm-11-05352-f004:**
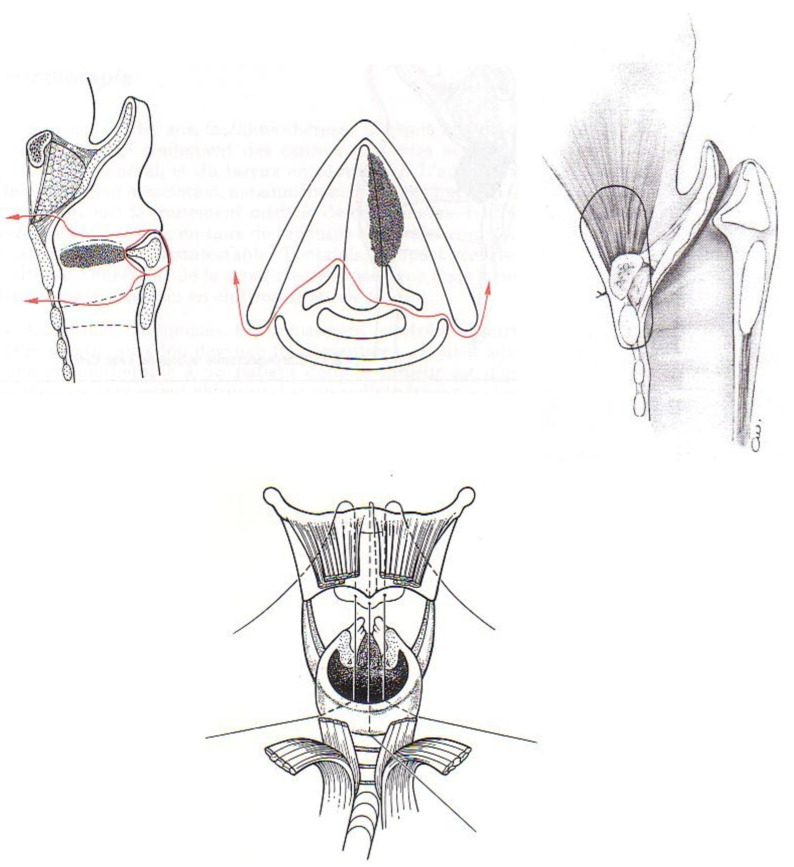
Supracricoid laryngectomy with crico-hyoido-epiglottopexy. Resection of the entire thyroid cartilage and both vocal folds with preservation of at least one arytenoid cartilage. Reconstruction: the closing of the larynx is carried out by an impaction between the cricoid cartilage, the epiglottis, the hyoid bone and the base of the tongue called a “crico-hyoido-epiglottopexy”.

## Data Availability

The study did not report any data.

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
