# Peer review of "Open Partial Laryngectomies: History of Laryngeal Cancer Surgery"

_jcm, 2022, doi:10.3390/jcm11185352_

Round 1

Reviewer 1 Report

This is an interesting historical review aiming at describing the history of open laryngeal cancer surgery. The first part of the historical review is adequate while the second part is partial and incomplete. First of all, the Authors cite the reference and the life of the Authors who described the technique but they do not include the exact date of publication of all techniques. The Authors cannot forget terms and acronyms like subtotal laryngectomy (SL), supracricoid partial laryngectomy (SCPL), supratracheal partial laryngectomy (STPL), cricohyoidopexy (CHP by Labayle cited by the Authors), tracheohyoidopexy (THP), tracheohyoidoepiglottopexy (THEP), partial glottic-subglottic laryngectomy (GLS), hemipharyngo-total laryngectomy (HPTL) by Bocca (also involved in the history of horizontal supraglottic laryngectomy and functional neck dissection). In this regard, they should include all the information with a chronological order. Although this paper has an historical perspective, old and new classification of open partial laryngectomy cannot be omitted or cited as they may help to better understand the transition from vertical to horizontal partial laryngectomies. The new OPHL classification from European Laryngological Society (ELS) and HOLS (Horizontal laryngectomy system) are nowadays part of this historical transition. The Authors included only three Figures (one concerning laryngoscopy and two concerning surgical techniques). Why? It would be appropriate to include a photo of any of the procedures to enhance the value of the review; on the contrary, it is better to remove all the Figures. Finally, the Authors did not include, among open laryngectomies, the technique of horizontal glottectomy by Calearo-Teatini. Why? Please, update the text and the references on this topic. The conclusion is very poor without any discussion; the Authors simply describe the limits of open partial laryngeal surgeries in favour of the results of more recent transoral laser microsurgery and transoral robotic surgery. Independently of the advancement of new techniques, open partial laryngectomies maintain a pivotal role not only in the management of primary laryngeal cancer but also in the salvage surgery of recurrent carcinoma; the Authors should comment on it in the conclusion.

Author Response

Reviewer 1:

This is an interesting historical review aiming at describing the history of open laryngeal cancer surgery. The first part of the historical review is adequate while the second part is partial and incomplete. First of all, the Authors cite the reference and the life of the Authors who described the technique but they do not include the exact date of publication of all techniques. The Authors cannot forget terms and acronyms like subtotal laryngectomy (SL), supracricoid partial laryngectomy (SCPL), supratracheal partial laryngectomy (STPL), cricohyoidopexy (CHP by Labayle cited by the Authors), tracheohyoidopexy (THP), tracheohyoidoepiglottopexy (THEP), partial glottic-subglottic laryngectomy (GLS), hemipharyngo-total laryngectomy (HPTL) by Bocca (also involved in the history of horizontal supraglottic laryngectomy and functional neck dissection).

We thank the reviewer for these comments.

We added Bocca's role in the international spread of supraglottic laryngectomy.

Bocca E. Supraglottic laryngectomy and functional neck dissection. J Laryngol Otol. 1966;80:831-8.

In this regard, they should include all the information with a chronological order. Although this paper has an historical perspective, old and new classification of open partial laryngectomy cannot be omitted or cited as they may help to better understand the transition from vertical to horizontal partial laryngectomies. The new OPHL classification from European Laryngological Society (ELS) and HOLS (Horizontal laryngectomy system) are nowadays part of this historical transition.

We thank the reviewer for these comments.

We added these sentences to the introduction: p.2, line 41: “The existing literature describes dozens of types of open partial laryngectomies. In our work, we choose to focus on a selection of publication that played an important role in illustrating two therapeutic extremes: transoral approach and total laryngectomy. We also provided historical examples of partial larytngectomies to support examples. We learned about these delicate surgeries in the "School of Laënnec" in contact with professors Henri Laccourreye, Daniel Brasnu and Ollivier Laccourreye. Finally, we used the European Laryngological Society (ELS) classification for open partial horizontal laryngectomies (OPHL) [4].”

The Authors included only three Figures (one concerning laryngoscopy and two concerning surgical techniques). Why? It would be appropriate to include a photo of any of the procedures to enhance the value of the review; on the contrary, it is better to remove all the Figures.

We do not have high-quality photo. However, we have a figure with the different partial open laryngectomies according to the ELS classification.
We included it as Figure 3.

Finally, the Authors did not include, among open laryngectomies, the technique of horizontal glottectomy by Calearo-Teatini. Why? Please, update the text and the references on this topic.

We thank the reviewer for these comments.

The existing literature describes dozens of types of open partial laryngectomies. In our work, we chose to focus on a selection of these that we believe still play a pivotal role in illustrating two therapeutical extremes: transoral approach on one hand or total laryngectomy on the other.

Precisely, we focused on OPL described in the ELS classification.

The approach developed by Calléaro-Teatini (1978) was the horizontal resection of the thyroid cartilage named “horizontal glottectomy” for glottic carcinoma involving the anterior commissure classified as cT1-T2. It allows easier and faster rehabilitation, even in elderly patients than CHEP.

« Among the many less common types of open partial laryngectomies, horizontal glottectomy, described in 1978 by Calearo and Teatini (who were both, by the way, strictly linked to our laryngological school), definitely represents a procedure that is nowadays quite limited in its application since it has been overshadowed, for the reported indications, by transoral laser microsurgery and radiotherapy » (cited by Succo G, Peretti G, Piazza C)

This is the reference of the types of OPL that we included.

Succo G, Peretti G, Piazza C. Reply to the comment to the article "Open partial horizontal laryngectomies: a proposal for classification by the working committee on nomenclature of the European Laryngological Society". Eur Arch Otorhinolaryngol. 2015 Apr;272(4):1043. doi: 10.1007/s00405-014-3228-7. Epub 2014 Aug 17. PMID: 25129373

The conclusion is very poor without any discussion; the Authors simply describe the limits of open partial laryngeal surgeries in favour of the results of more recent transoral laser microsurgery and transoral robotic surgery. Independently of the advancement of new techniques, open partial laryngectomies maintain a pivotal role not only in the management of primary laryngeal cancer but also in the salvage surgery of recurrent carcinoma; the Authors should comment on it in the conclusion.

We thank the reviewer for these comments.

According to the comment of the reviewer, we added two paragraphs:

5.4. Evolution of partial laryngectomies over the last thirty years

5.5. Current indications for partial laryngectomies

p.9, line 220: “5.4. Evolution of partial laryngectomies over the last thirty years

In the past thirty years, there has been a major transformation in the way we treat LSCC, including a decline in the use of open surgery as first-line of treatment for a certain proportion of these tumors [48-50]. This evolution was made possible by several factors.

First, the LSCC's incidence has decreased in most developed countries, partially, as a result of public health agencies' efforts to decrease tobacco consumption. Second, advances in chemotherapy and radiation therapy (RT) have led to highly effective non-surgical regimens for patients with advanced laryngeal cancers, with the added advantage of laryngeal preservation in many cases. The Veterans Administration Study published in 1991 established the fact that the response to neoadjuvant CT-scan predicts the response of a tumor to RT. Patients with advanced tumors that responded either partially or completely to CT were treated with RT, and total laryngectomy was reserved for non-responders. This made it possible to preserve the larynx in a significant number of patients with locally advanced laryngeal cancer, while achieving local control and overall survival results equivalent to those achieved with initial total laryngectomy. By 2003, results of the RTOG 93-11 trial, utilizing CCRT as initial treatment, were published, demonstrating a higher rate of laryngeal preservation with this protocol. Surgery was reserved for treatment failures. This concept changed the paradigm for management of advanced laryngeal cancer, greatly reducing the number of performed laryngectomies. While partial supracricoid laryngectomy has been employed for selected patients, total laryngectomy is the usual procedure for salvage or failure after non surgical treatment.

Third, technological advances with the widespread availability of operating microscopes, endoscopes, lasers, image-guided surgery and more recently robotics, are transforming our surgical approaches, with transoral minimally invasive techniques greatly improving the postoperative course and functional outcomes for selected tumors (see Laryngeal Cancer Surgery-Part II).”

p.9, line 240: “5.5. Current indications for partial laryngectomies

From our experience and recently published articles, the current indications for partial laryngectomies are laryngeal tumors with inadequate transoral exposure, certain tumors of the anterior commissure with vertical development (see Laryngeal Cancer Surgery-Part II) and selected salvage LSCC, after radiation therapy failure. However, experience of many contemporary surgeons with partial laryngectomy is quite limited. For the treatment of localized RT failure laryngeal cancer, the surgeon must be perfectly familiar with the type of extension of the tumor as well as the indications for partial laryngectomies. Nevertheless, partial laryngectomy should be used with caution in patients requiring salvage surgical therapy for recurrent or persistent laryngeal tumor. Recurrences after RT tend to be submucosal and difficult to evaluate. The only type of partial laryngectomies reported in the literature are the supracricoid partial laryngectomies [51,52]. In a systematic review, De Virgilio reported eleven papers (251 patients from 1990 to December 2017) with 2-year local control and 5-year overall survival were 92 and 70% respectively. The larynx preservation rate was 85.2%. The decannulation rate was 92.1%, and the swallowing recovery was 96.5% (PEG dependance and the aspiration pneumonia rate were 3.5 and 6.4%, respectively) [52].”

General comment: A Native speaker UK proofread the paper, all changes are in green

Reviewer 2 Report

Well developed job. It is necessary to know the history of our specialty.

1.- The importance of knowing the history of a surgical procedure. Described for more than 200 years. With the technical improvements that have been incorporated, it should be a procedure known to any head and neck surgeon

2.- it is a historical review, but it is very interesting from my point of view.

3.- The originality is based on the exhaustive bibliographic review carried out. Consulting classic papers, some of them difficult to find.

4.- Groups in a single work all the references on partial laryngeal surgery in a single paper

5- I am not a linguistic expert, but I consider that it is sufficiently well written and that its reading is not complex and easy to understand.

6.- The conclusions agree. It is very difficult to summarize a historical review in an abstract or conclusion.

Unlike other types of work, it is necessary to read the entire paper to know all the work done.

7.- they address the main question.

Author Response

Reviewer 2:

Well developed job. It is necessary to know the history of our specialty.

1.- The importance of knowing the history of a surgical procedure. Described for more than 200 years. With the technical improvements that have been incorporated, it should be a procedure known to any head and neck surgeon

2.- it is a historical review, but it is very interesting from my point of view.

3.- The originality is based on the exhaustive bibliographic review carried out. Consulting classic papers, some of them difficult to find.

4.- Groups in a single work all the references on partial laryngeal surgery in a single paper

5- I am not a linguistic expert, but I consider that it is sufficiently well written and that its reading is not complex and easy to understand.

6.- The conclusions agree. It is very difficult to summarize a historical review in an abstract or conclusion.

Unlike other types of work, it is necessary to read the entire paper to know all the work done.

7.- they address the main question.

We are very grateful to reviewer 2 for these kind comments.

Reviewer 3 Report

The information presented in this article is interesting, succinct, and well-organized – as a surgeon, I very much enjoyed reading this short review about the history of partial laryngectomy. However, the manuscript suffers from grammatical issues that limit its readability. While the text is understandable, I would highly recommend the authors have a native English writer review the manuscript. There are several syntactical errors and odd wording choices that are beyond the scope of a peer review to address. Additionally, this historical review misses much of the more recent history regarding open partial laryngectomy and should have at least some text regarding the past 50 years of changes to the procedure.

Abstract

1.       Please specify LSCC to laryngeal squamous cell carcinoma (LSCC) in the first reference

2.       There are stylistic issues throughout the abstract but one main issue is use of an article (“the”) when referring to surgeries or techniques. I would change all to “partial laryngeal surgery” rather than “the partial laryngeal surgery.” Also remove “the” from “The general anesthesia…”

3.       Change “The partial laryngectomy raised importance in the medical community in 1888…” to “The partial laryngectomy was brought to importance in the medical community in 1888

4.       This sentence is very confusing: “The origin of its dead led to debates and reflections about the importance of semiology, biopsy and the early diagnosis of laryngeal cancer to propose partial laryngectomy.” Either delete or specify how Frederick III’s death influence laryngectomy care. Also Frederick III should be referred by the pronouns “he” or “him” rather than “it.”

5.       Change “The check-up of” to “surveillance”

The Pioneers

6.       Change “exeresis” to “extirpation”

The Prince of Prussia

7.       I’m not sure what “Semiology” means in this context. Are the authors referring to staging?

8.       Is there a typo relating to the overall survival numbers: “the 3-year death rate and overall survival of midline thyrotomies were 8% and 44.7%, respectively, while the total laryngectomy was associated with 3-year death rate and overall survival of 8.7% and 5.85%” How can one be 44.7% and the other be 5.85%?

First part of the 20th century

9.       Would it be possible to include a figure of a vertical partial laryngectomy? This is arguably as important as horizontal partial laryngectomies for open laryngeal conservation surgery and an accompanying figure would be beneficial to the text.

Second part of the 20th century

10.   The article jumps from the mid-1950s to present day very quickly. This misses a large and critical time period for open partial laryngectomy. There are several important topics to address:

a.       Voice outcomes from these procedures and how they have evolved.

b.       Declining rates of open partial laryngectomy due to TLM and chemoradiation. Many head and neck surgeons no longer perform or receive training in these approaches – what are the reasons for their decline?

c.       Current advances in open approaches. There are several papers about combined robotic-open partial laryngectomies that should be included (for example “Hybrid supracricoid partial laryngectomy with cricohyoidoepiglottopexy via transoral robotic surgery” by Nakayama, Holsinger, and Orosco).

Author Response

The information presented in this article is interesting, succinct, and well-organized – as a surgeon, I very much enjoyed reading this short review about the history of partial laryngectomy. However, the manuscript suffers from grammatical issues that limit its readability. While the text is understandable, I would highly recommend the authors have a native English writer review the manuscript. There are several syntactical errors and odd wording choices that are beyond the scope of a peer review to address. Additionally, this historical review misses much of the more recent history regarding open partial laryngectomy and should have at least some text regarding the past 50 years of changes to the procedure.

We are very grateful to reviewer 3 for these kind comments.

We added two paragraphs:

5.4. Evolution of partial laryngectomies over the last thirty years

5.5. Current indications for partial laryngectomies

p.9, line 220: “5.4. Evolution of partial laryngectomies over the last thirty years

In the past thirty years, there has been a major transformation in the way we treat LSCC, including a decline in the use of open surgery as first-line of treatment for a certain proportion of these tumors [48-50]. This evolution was made possible by several factors.

First, the LSCC's incidence has decreased in most developed countries, partially, as a result of public health agencies' efforts to decrease tobacco consumption. Second, advances in chemotherapy and radiation therapy (RT) have led to highly effective non-surgical regimens for patients with advanced laryngeal cancers, with the added advantage of laryngeal preservation in many cases. The Veterans Administration Study published in 1991 established the fact that the response to neoadjuvant CT-scan predicts the response of a tumor to RT. Patients with advanced tumors that responded either partially or completely to CT were treated with RT, and total laryngectomy was reserved for non-responders. This made it possible to preserve the larynx in a significant number of patients with locally advanced laryngeal cancer, while achieving local control and overall survival results equivalent to those achieved with initial total laryngectomy. By 2003, results of the RTOG 93-11 trial, utilizing CCRT as initial treatment, were published, demonstrating a higher rate of laryngeal preservation with this protocol. Surgery was reserved for treatment failures. This concept changed the paradigm for management of advanced laryngeal cancer, greatly reducing the number of performed laryngectomies. While partial supracricoid laryngectomy has been employed for selected patients, total laryngectomy is the usual procedure for salvage or failure after non surgical treatment.

Third, technological advances with the widespread availability of operating microscopes, endoscopes, lasers, image-guided surgery and more recently robotics, are transforming our surgical approaches, with transoral minimally invasive techniques greatly improving the postoperative course and functional outcomes for selected tumors (see Laryngeal Cancer Surgery-Part II).”

p.9, line 240: “5.5. Current indications for partial laryngectomies

From our experience and recently published articles, the current indications for partial laryngectomies are laryngeal tumors with inadequate transoral exposure, certain tumors of the anterior commissure with vertical development (see Laryngeal Cancer Surgery-Part II) and selected salvage LSCC, after radiation therapy failure. However, experience of many contemporary surgeons with partial laryngectomy is quite limited. For the treatment of localized RT failure laryngeal cancer, the surgeon must be perfectly familiar with the type of extension of the tumor as well as the indications for partial laryngectomies. Nevertheless, partial laryngectomy should be used with caution in patients requiring salvage surgical therapy for recurrent or persistent laryngeal tumor. Recurrences after RT tend to be submucosal and difficult to evaluate. The only type of partial laryngectomies reported in the literature are the supracricoid partial laryngectomies [51,52]. In a systematic review, De Virgilio reported eleven papers (251 patients from 1990 to December 2017) with 2-year local control and 5-year overall survival were 92 and 70% respectively. The larynx preservation rate was 85.2%. The decannulation rate was 92.1%, and the swallowing recovery was 96.5% (PEG dependance and the aspiration pneumonia rate were 3.5 and 6.4%, respectively) [52].”

Abstract

  1. Please specify LSCC to laryngeal squamous cell carcinoma (LSCC) in the first reference

We have modified

  1. There are stylistic issues throughout the abstract but one main issue is use of an article (“the”) when referring to surgeries or techniques. I would change all to “partial laryngeal surgery” rather than “the partial laryngeal surgery.” Also remove “the” from “The general anesthesia…”

We have modified

  1. Change “The partial laryngectomy raised importance in the medical community in 1888…” to “The partial laryngectomy was brought to importance in the medical community in 1888

We have modified

  1. This sentence is very confusing: “The origin of its dead led to debates and reflections about the importance of semiology, biopsy and the early diagnosis of laryngeal cancer to propose partial laryngectomy.” Either delete or specify how Frederick III’s death influence laryngectomy care. Also Frederick III should be referred by the pronouns “he” or “him” rather than “it.”

We have modified

  1. Change “The check-up of” to “surveillance”

We have modified

The Pioneers

  1. Change “exeresis” to “extirpation”

We have modified

The Prince of Prussia

  1. I’m not sure what “Semiology” means in this context. Are the authors referring to staging?

In our opinion, "Semiology" can be defined as the study of signs (dysphonia, pain..). However, we changed the term with “semiotics”

  1. Is there a typo relating to the overall survival numbers: “the 3-year death rate and overall survival of midline thyrotomies were 8% and 44.7%, respectively, while the total laryngectomy was associated with 3-year death rate and overall survival of 8.7% and 5.85%” How can one be 44.7% and the other be 5.85%?

There is indeed an error. We specified the number of laryngectomies. In this study, the author reported that the post-operative mortality rate and 3-year overall survival of midline thyrotomies (n=88) were 9.8% and 8.7%, respectively, while total laryngectomy (n = 188) was associated with a postoperative mortality rate and overall survival of 44.7% and 5.85%, respectively.

First part of the 20thcentury

  1. Would it be possible to include a figure of a vertical partial laryngectomy? This is arguably as important as horizontal partial laryngectomies for open laryngeal conservation surgery and an accompanying figure would be beneficial to the text.

We have a figure with the different partial open laryngectomies according to the ELS classification.
We included it as Figure 3.

Second part of the 20thcentury

  1. The article jumps from the mid-1950s to present day very quickly. This misses a large and critical time period for open partial laryngectomy. There are several important topics to address:
  2. Voice outcomes from these procedures and how they have evolved.

We thank the reviewer for these comments. It is a very broad subject.

We included: p.9, line 208: “Many studies have demonstrated that vocal quality after partial laryngectomy is related to the extent of the resection and the reconstruction methods. The spoken voice is called "neoglottic substitution voice" after partial vertical laryngectomy (43, 44) and "neolaryngeal substitution voice" after supracricoid laryngectomy (OPHL type IIa/IIb) (45-47). For open partial laryngectomies, the voice characteristics will depend on i) the shape and nature of the remaining structures and the amount of effort and adaptation to reach a vibrating neoglottis / neolaryngeal voicing sphincter, ii) collateral constraints of external surgeries such as a transient tracheostomy or nasogastric tube that modify and lengthen the dynamics of vocal recovery; For endoscopic surgeries, the physiological shape of the phonatory glottis is not modified.Thus, the vocal rehabilitation aims to improve the closure of the neoglottis and the quality of the mucosal vibration with voice tone improvement (see Laryngeal Cancer Surgery-Part II). After partial laryngectomy, voice rehabilitation is long and requires the patient’s effort. In the case of vertical partial laryngectomies, many techniques (flap, etc.) have been developed to improve the voice (43,44). According to our experience and from literature data, vocal progress is achievable even several months after partial laryngectomy (46,47).”

  1. Declining rates of open partial laryngectomy due to TLM and chemoradiation. Many head and neck surgeons no longer perform or receive training in these approaches – what are the reasons for their decline?

We thank the reviewer for these comments.

We added two paragraphs:

5.4. Evolution of partial laryngectomies over the last thirty years

5.5. Current indications for partial laryngectomies

p.9, line 220: “5.4. Evolution of partial laryngectomies over the last thirty years

In the past thirty years, there has been a major transformation in the way we treat LSCC, including a decline in the use of open surgery as first-line of treatment for a certain proportion of these tumors [48-50]. This evolution was made possible by several factors.

First, the LSCC's incidence has decreased in most developed countries, partially, as a result of public health agencies' efforts to decrease tobacco consumption. Second, advances in chemotherapy and radiation therapy (RT) have led to highly effective non-surgical regimens for patients with advanced laryngeal cancers, with the added advantage of laryngeal preservation in many cases. The Veterans Administration Study published in 1991 established the fact that the response to neoadjuvant CT-scan predicts the response of a tumor to RT. Patients with advanced tumors that responded either partially or completely to CT were treated with RT, and total laryngectomy was reserved for non-responders. This made it possible to preserve the larynx in a significant number of patients with locally advanced laryngeal cancer, while achieving local control and overall survival results equivalent to those achieved with initial total laryngectomy. By 2003, results of the RTOG 93-11 trial, utilizing CCRT as initial treatment, were published, demonstrating a higher rate of laryngeal preservation with this protocol. Surgery was reserved for treatment failures. This concept changed the paradigm for management of advanced laryngeal cancer, greatly reducing the number of performed laryngectomies. While partial supracricoid laryngectomy has been employed for selected patients, total laryngectomy is the usual procedure for salvage or failure after non surgical treatment.

Third, technological advances with the widespread availability of operating microscopes, endoscopes, lasers, image-guided surgery and more recently robotics, are transforming our surgical approaches, with transoral minimally invasive techniques greatly improving the postoperative course and functional outcomes for selected tumors (see Laryngeal Cancer Surgery-Part II).”

p.9, line 240: “5.5. Current indications for partial laryngectomies

From our experience and recently published articles, the current indications for partial laryngectomies are laryngeal tumors with inadequate transoral exposure, certain tumors of the anterior commissure with vertical development (see Laryngeal Cancer Surgery-Part II) and selected salvage LSCC, after radiation therapy failure. However, experience of many contemporary surgeons with partial laryngectomy is quite limited. For the treatment of localized RT failure laryngeal cancer, the surgeon must be perfectly familiar with the type of extension of the tumor as well as the indications for partial laryngectomies. Nevertheless, partial laryngectomy should be used with caution in patients requiring salvage surgical therapy for recurrent or persistent laryngeal tumor. Recurrences after RT tend to be submucosal and difficult to evaluate. The only type of partial laryngectomies reported in the literature are the supracricoid partial laryngectomies [51,52]. In a systematic review, De Virgilio reported eleven papers (251 patients from 1990 to December 2017) with 2-year local control and 5-year overall survival were 92 and 70% respectively. The larynx preservation rate was 85.2%. The decannulation rate was 92.1%, and the swallowing recovery was 96.5% (PEG dependance and the aspiration pneumonia rate were 3.5 and 6.4%, respectively) [52].”

  1. Current advances in open approaches. There are several papers about combined robotic-open partial laryngectomies that should be included (for example “Hybrid supracricoid partial laryngectomy with cricohyoidoepiglottopexy via transoral robotic surgery” by Nakayama, Holsinger, and Orosco).

We thank the reviewer for these comments.

TLM and TORS are developed in Laryngeal Cancer Surgery-Part II (a second paper in revision).

Round 2

Reviewer 3 Report

I am appreciative to the authors for making major revisions and responding to all of the reviewer comments. The additional figures and text make this a comprehensive and interesting review of the history of partial laryngectomy.